# The Effects of Core Stabilization Exercise with the Abdominal Drawing-in Maneuver Technique versus General Strengthening Exercise on Lumbar Segmental Motion in Patients with Clinical Lumbar Instability: A Randomized Controlled Trial with 12-Month Follow-Up

**DOI:** 10.3390/ijerph18157811

**Published:** 2021-07-23

**Authors:** Rungthip Puntumetakul, Pongsatorn Saiklang, Weerasak Tapanya, Thiwaphon Chatprem, Jaturat Kanpittaya, Preeda Arayawichanon, Rose Boucaut

**Affiliations:** 1Research Center of Back, Neck, Other Joint Pain and Human Performance (BNOJPH), Khon Kaen University, Khon Kaen 40002, Thailand; Thiwaphon.ao@gmail.com; 2Department of Physical Therapy, Faculty of Associated Medical Sciences, Khon Kaen University, Khon Kaen 40002, Thailand; 3Faculty of Physical Therapy, Srinakharinwirot University, Nakhon Nayok 26120, Thailand; Pongsatornsa@g.swu.ac.th; 4Department of Physical Therapy, School of Allied Health Sciences, University of Phayao, Phayao 56000, Thailand; Weerasak.ta@up.ac.th; 5Department of Radiology, Faculty of Medicine, Khon Kaen University, Khon Kaen 40002, Thailand; Jatkan@kku.ac.th; 6Department of Rehabilitation Medicine, Faculty of Medicine, Khon Kaen University, Khon Kaen 40002, Thailand; Prearr@yahoo.com; 7iCAHE (International Centre for Allied Health Evidence), School of Health Sciences (Physiotherapy), University of South Australia, Adelaide, SA 5001, Australia; Rose.Boucaut@unisa.edu.au

**Keywords:** low back pain, stability exercises, radiography, lumbar translation, lumbar rotation, electromyography

## Abstract

Trunk stability exercises that focus on either deep or superficial muscles might produce different effects on lumbar segmental motion. This study compared outcomes in 34 lumbar instability patients in two exercises at 10 weeks and 12 months follow up. Participants were divided into either Core stabilization (deep) exercise, incorporating abdominal drawing-in maneuver technique (CSE with ADIM), or General strengthening (superficial) exercise (STE). Outcome measures were pain, muscle activation, and lumbar segmental motion. Participants in CSE with ADIM had significantly less pain than those in STE at 10 weeks. They showed significantly more improvement of abdominal muscle activity ratio than participants in STE at 10 weeks and 12 months follow-up. Participants in CSE with ADIM had significantly reduced sagittal translation at L4-L5 and L5-S1 compared with STE at 10 weeks. Participants in CSE with ADIM had significantly reduced sagittal translations at L4-L5 and L5-S1 compared with participants in STE at 10 weeks, whereas STE demonstrated significantly increased sagittal rotation at L4-L5. However, at 12 months follow-up, levels of lumbar sagittal translation were increased in both groups. CSE with ADIM which focuses on increasing deep trunk muscle activity can reduce lumbar segmental translation and should be recommended for lumbar instability.

## 1. Introduction

Chronic low back pain (CLBP) is the most common musculoskeletal disorder worldwide, and it causes reduced physical performance, psychosocial problems, and economic burden [1,2,3]. Clinical lumbar instability (CLI) is defined as the inability of the spine to maintain its normal patterns of displacement under physiologic loads; there is no initial or additional neurologic deficit, no major deformity, and no incapacitating pain [4]. The prevalence of CLI is approximately 13–35% among patients suffering from CLBP [5,6,7,8]. Progressive pathology in CLI may induce spondylolisthesis, which can present a high risk of neurological problems such as cauda equina syndrome [9,10]. In the last two decades, surgical spinal fusion, often aimed at reducing instability, has increased about 2.4 times [11]. However, spinal fusion is not always successful; it may present complications such as infection, failed back syndrome, and psychological problems [12,13,14]. Furthermore, 22% of patients who have undergone spinal operations have returned with persistent lumbar instability three years after surgery [15].

Lumbar segmental motion is related to trunk muscle activation and may explain lumbar segmental pathologies [4,16,17]. Trunk stability exercises have been recommended for patients with CLI to decrease pain, reduce disability, improve quality of life, trunk muscle function, and quality of lumbar segmental motion [5,18,19,20,21,22,23]. Core stabilization exercise with the abdominal drawing-in maneuver technique (CSE with ADIM) has been recommended for CLBP patients with CLI as it specifically improves neuromuscular control of deep trunk muscles: transversus abdominis (TrA) and lumbar multifidus (LM) [18,21,24,25]. These muscles play an important role in lumbar segmental stabilization [4,26]. CSE with ADIM activates the deep trunk muscles and is recommended for people who suffer from spinal instability because it enhances deep trunk muscle function and neuromuscular control that may increase segmental lumbar stability, thereby preventing excessive lumbar segmental motion.

In a previous study, Javadian et al. in 2015 found that patients with CLI who received CSE with ADIM combined with general strengthening exercise (STE), rather than STE alone, showed reduced excess lumbar vertebral translation and rotation. They indicated that both exercise groups (CSE with ADIM, and STE) were effective, but CSE with ADIM had additional benefits that were statistically significant after eight weeks of the exercises [22]. However, their study did not assess the neuromuscular system.

Electromyography or EMG is a quantitative analysis method used to measure either the coordination of muscle activation [27] or muscle activation [18,28,29]. Hubley-Kozey and Vezina in 2002 investigated the effects of three exercises (pelvic-tilt, CSE with ADIM, and level 1 of the trunk stability test) on trunk stability by focusing on the coordination of trunk muscles in patients with low back pain. They demonstrated that low levels of muscle activation may produce positive outcomes, and suggested this was important, rather than only focusing on the amount and ratios of muscle activation. Their result does not reflect the outcome of the prolonged effect of exercise, it represents the muscle activated at the time of exercise performance. Therefore, the effect of long-term muscle activation via exercise still requires investigation [27].

Puntumetakul et al. in 2013 and Areeudomwong et al. in 2012 reported that CSE with ADIM can improve trunk muscle activation in terms of increased deep trunk muscle and decreased superficial trunk muscle activation when participants received 10-weeks of exercise intervention. The findings supported the hypothesis related to trunk muscle recruitment patterns in patients with clinical lumbar instability. After 10 weeks of CSE training, the activation ratio of the abdominal muscles (TrA and IO/RA) was significantly increased, with higher activation of TrA and IO muscles. However, their intervention assessed only muscle activation with a 3-month follow-up and did not assess lumbar segmental motion [18].

To date, no research has examined the long-term effect of CSE with ADIM on lumbar segmental motion together with trunk muscle activity to see whether the results of lumbar segmental motion and muscle activation would be in the direction we expected or not. That is, the higher the deep trunk muscle activation, the lower the lumbar segmental motion. We decided to use EMG to measure muscle activation which is a similar method to that used in our previous studies [18,29]. This is the first study to examine both muscle activity and lumbar segmental motion (X-ray) variables in patients with CLI. The current study aimed to compare the effect of 10 weeks of CSE with ADIM vs. STE and at 12 months follow-up, on lumbar segmental motion (L3-L4, L4-L5, and L5-S1), trunk muscle activation, and pain intensity in CLBP patients with CLI. This study hypothesized that lumbar segmental motion in the CES with ADIM group would reduce more, at the 10-week and 12-month follow-up period, than in the STE groups.

## 2. Materials and Methods

### 2.1. Design and Procedure

An assessor-blinded randomized controlled trial was conducted with three measurement points, including at baseline, at the end of the 10-week exercise program, and at 12-month follow-up. The study was approved by the Ethics Committee for Human Research (HE562257) at Khon Kaen University, based on the Declaration of Helsinki. This research protocol was approved in ClinicalTrials.gov (ID number: NCT02895828) by the United States National Library of Medicine (NLM). The study was conducted in the Musculoskeletal Physical Therapy Laboratory between August 2016 and August 2018 at Khon Kaen University, Thailand.

### 2.2. Participants

Males and females aged 20–60 years with low back pain of at least three months duration were recruited from the Department of Physical Medical and Rehabilitation, outpatients of Srinagarind Hospital, and the Physical Therapy Clinic in the Faculty of Associated Medical Sciences, Khon Kaen University, Thailand. Participants completed a 13-item subjective examination and undertook three clinical tests. These were tests for aberrant movement signs, painful catch signs, and prone instability. Positive test results on two out of the three tests led to a diagnosis of lumbar instability by the physical therapist, who had 30 years of experience (Researcher RP) [18,20,25]. Patients with other spinal disorders were excluded from the study by a physiatrist (Researcher PA), for example, disc herniation, spondylolisthesis, spinal deformities, spinal infection, pregnancy, back muscle strain, and excessive lumbar segmental motion [4]. The process of lumbar segmental motion measurement was assessed by a radiologist (Researcher JK), using flexion-extension radiographs according to criteria proposed by Panjabi in 2003 [4]. In addition, participants who regularly performed CSE with ADIM or STE were excluded. Participants provided informed consent prior to their involvement in the study. The flow of progress throughout the study is presented in Figure 1.

Sixty-five people with CLBP responded to the recruitment advertisement. Thirty-one participants who did not meet the inclusion criteria were excluded. These participants were excluded for a number of reasons, including 2 participants with disc herniation, three participants with spondylolisthesis, and three participants with back muscle strain. Other participants excluded showed a positive test result on only one test, including eight participants with positive painful catch signs, five participants with positive prone instability tests, three participants with aberrant movement signs positive, and seven participants having positive results out of three tests. The other 34 eligible participants entered the study. The participants were randomized into the CSE with ADIM group (*n* = 17) or the STE group (*n* = 17).

All participants completed 10 weeks of exercises and the 12-month assessment follow-up. Some participants reported non-serious adverse effects (e.g., the discomfort of the low back, general fatigue) and all were able to continue their exercise activity throughout the program. There was no significant difference in the participant characteristics shown between the two groups (Table 1).

### 2.3. Sample Size Determination

The sample size, using a formula of difference between two independent groups, was calculated from the difference in the mean and standard deviation values between groups of L4-5 segmental translation variable at follow-up after 12 months of the exercises (2.33 ± 0.96 mm and 3.11 ± 0.80 mm of CSE with ADIM and STE group, respectively) in 10 participants (CSE with ADIM = 5, STE = 5) obtained from a pilot study, assuming 80% power and a 20% attrition rate. An optimal sample size per group was calculated to be at least 17 individuals, which achieved an effect size of 0.88.

### 2.4. Randomization

The randomization sequence was computer-generated with gender stratification, with two-, four-, and six-block sizes. Researcher WT, who undertook the randomization process, was not involved in participant recruitment or exercise prescription. The results of randomization were concealed from participants in an opaque envelope with consecutive numbering.

### 2.5. Exercise Intervention

Eligible participants were enrolled in this study and randomly assigned to one of two exercise programs, either CSE with ADIM or STE. Researchers TC and PS, who were blinded to the outcome measurements, were assigned to demonstrate and supervise each of the 10-week exercise programs. Participants were trained face-to-face in their exercise by the physical therapist. Both programs participated in 20-min sessions, twice a week for 10 weeks in the laboratory. Participants in both groups were asked to practice the exercise daily at home using the same stage, position, and repetition as in the laboratory, and they were asked to record details of this in a logbook. Additionally, the research assistant made a phone call to the participants to motivate them with their daily home exercise. The researchers monitored and made decisions about the progression of the exercises at every session for each participant based on the correct performance of the previous exercise stage. If any participants failed to accurately perform the previous exercise, they were retrained until they succeeded, prior to progressing to a more advanced exercise.

#### 2.5.1. Core Stabilization Exercise with Abdominal Drawing-in Maneuver Technique (CSE with ADIM)

Details of the CSE with the ADIM program have been described in a previous study [18] (Appendix A). Participants in the CSE with the ADIM group were instructed to perform the CSE with the ADIM program by a physical therapist (Researcher PS) in a three-phased approach. On the first day, before starting the exercise program, participants were instructed on how to perform the CSE with ADIM by using a pressure biofeedback device (Chattanooga Australia Pty Ltd., Brisbane, QLD, Australia), which provided visual feedback and enhanced exercise performance. The pressure biofeedback device was placed under the lower abdomen with the lower edge in line with the anterior superior iliac spine and inflated to 70 mm of mercury [30,31]. Optimal performance of the test reduces the pressure by approximately 4–10 mm of mercury in the absence of spinal or pelvic movement [31]; electromyographic biofeedback (MP100, BIOPAC Systems Inc., Goleta, CA, USA) was used to facilitate correct performance during the practice session. When all participants could perform CSE with ADIM accurately using the pressure biofeedback device, they were then ready and were allowed to start the first phase of the exercise program. The first “cognitive stage” focused on correctly isolating low-load activation of the TrA and LM muscles using the ADIM technique in minimal loading positions of prone lying and sitting (in the 1st–2nd week). The second phase was the “associative stage”, where participants applied controlled upper or/and lower extremity movement together with controlled TrA and LM activity. This phase was performed in the 3rd–7th week. It was started as a progressive exercise after the participant could correctly control TrA and LM contraction. The third phase, or the “autonomous stage”, occurred during weeks 8–10th. In this stage, participants were trained to maintain co-contraction of TrA and LM while walking and performing a chosen task that they reported as aggravating their pain.

The exercise progression each week was evaluated by researcher PS. Participants were asked to perform their previous exercise. If the participants had pain and did not correctly perform the exercises, they were re-trained in the previous exercise until they could do it. However, if they had no pain and successfully performed the previous exercise, they then progressed to the next position of the exercise. Each exercise session lasted 20 min.

#### 2.5.2. General Strengthening Exercise (STE)

For the STE group, participants were instructed by a physical therapist (Researcher TC) to activate their superficial trunk muscles, erector spinae (ES), and rectus abdominis (RA) muscles. The STE program activated these muscles through body extension ES and flexion RA [25,32,33]. We modified the general strengthening trunk exercise described by Koumantakis et al. in 2005 [32] by extending the duration of the exercise program to 10 weeks (Appendix A). Each pose of the program was repeated 10 times, with 10-s holds, and 5 min was set for a resting interval between poses.

### 2.6. Outcome Measurements

#### 2.6.1. Pain Intensity

The average pain within a 24-h period was measured using an 11-point numeric rating scale (NRS) (range 0–10), where 0 represents no pain and 10 represents the worst possible pain [34,35]. Participants were asked to indicate the number that represented their pain over the past 24 h. Pain intensity was measured by researcher RP.

#### 2.6.2. Trunk Muscle Activity

Four-channel surface electromyography (sEMG) was used to continuously record EMGs (BioPacTM MP36). The sampling rate was 1000 Hz, with a signal amplification of gain ×1000 and a common-mode rejection ratio (CMRR) of 110 dB. The signal frequency was band-pass filtered between 10 and 500 Hz. A personal computer with an A/D converter was used to record and analyze the EMG data. The raw data generated during sub-maximal voluntary isometric contraction (sub-MVIC) was used instead of MVIC since sub-MVIC is better able to detect changes in levels of motor activity during the performance of low load tasks than MVIC [36].

The skin over the boundaries of the right side of the transversus abdominis (TrA) and internal oblique (IO) (TrA & IO), rectus abdominis (RA), lumbar multifidus (LM), and iliocostalis lumborum pars thoracis (ICLT) muscles were prepared to reduce skin impedance to less than 5 kΩ by shaving hair at the electrode sites, cleaning the sites with alcohol, and abrading the skin with fine sandpaper. After that, pairs of active disc disposable Ag/AgCl surface electrodes (EL 503) with electrical contact surface areas of 1 cm^2^ and a center-to-center spacing of 2.5 cm were attached parallel to each muscle on the right side, as previously reported.

For the TrA & IO muscle, electrodes were placed horizontally at 2 cm inferomedially to the anterior superior iliac spine [37]. For the RA muscle, electrodes were placed in the superior-inferior direction on its muscle belly at a 3 cm lateral distance from the navel [38]. For the LM muscle, electrodes were placed at the level of L5, parallel to a line connecting the posterior superior iliac spine and L1-L2 interspinous space [39]. For the ICLT muscle, electrodes were placed in the cephalo-caudal direction at a level of L1 of the spinous process, midway between the midline and lateral aspect of the body [40]. The two ground electrodes of the right RA and ICLT muscles were placed over the ribcage, and the other two ground electrodes of the right TrA & IO and LM muscles were placed over the right iliac crest. Snap leads were used to connect the surface electrodes with the amplifier to transfer signals.

The ADIM task was used for representing all muscle activities. It was separately performed before and after 10-week of exercise program. The participants were in a 4-point kneeling position with a small pillow placed under their ankles. Participants were asked to perform the ADIM technique task with a 10-s hold, three times with a 2-min rest between trials [24,29].

Normalization of abdominal and back muscle activities was performed using sub-maximal voluntary isometric contractions. For the abdominal muscles, participants raised both feet 1 cm from the floor in a crook lying position with a 5-s hold [36,41,42,43]. For the back muscles, participants performed a double knee raise 5 cm from the floor in the prone lying position with a 5-s hold [36,42,44]. The root mean square (RMS) values of the middle 3 s of the 5 s testing period were analyzed [29,36,42,44]. The RMS values of EMG signals produced during an isometric hold of these activities were then used to normalize the sEMG signals obtained during the ADIM task. All the normalized RMS values were expressed as a percentage of sub-MVIC (%sub-MVIC). Trunk muscle activity was measured by researcher RP.

#### 2.6.3. Trunk Muscle Ratio

The activation ratio of the abdominal (TrA and IO/RA) and back (LM/ICLT) muscles was calculated using the normalized data [18,29]. The ratio muscle activation of mean RMS value (mean RMS of deep muscle divided by mean RMS of superficial muscle) has previously been used to represent the change in muscle activation [45,46]. For example, when the ratio activation of the deep trunk muscles relative to the superficial trunk muscle is increased, shifts in the activation patterns of the muscles are represented with a higher activation level of the deep trunk muscles and a lower activation level of the superficial trunk muscle. The ratio activation of the trunk muscles may be an important indicator of motor control patterns [46].

#### 2.6.4. Lumbar Segmental Motion

The primary outcome of the study was lumbar segmental motion, which was measured by flexion and extension X-ray films. During side-lying flexion-extension, sagittal translation and rotation were emphasized to define lumbar instability. This position was used to avoid the influence of muscle bracing and aggravating the participant’s pain during standing movement, full flexion-extension in a side-lying position for the X-ray assessment was applied in the current study [47,48]. The translation and rotation drawing measurement lines and calculation formula are illustrated in Figure 2. Angles of measurement are formed by the intersection of two lines drawn along the inferior endplate of the upper vertebra and the superior endplate of the lower vertebra. The difference in intervertebral angles between flexion and extension is defined as the amount of rotation (degrees). For translation measurement (Figure 2), lines were drawn to bisect the endplate angle. The difference between the two distances from (A) and (B) defined the amount of translation (mm) [49]. The process of lumbar segmental motion measurement was assessed by a radiologist (Researcher JK), using flexion-extension radiographs according to criteria proposed by Panjabi in 2003 [4].

### 2.7. Statistical Analyses

Data are presented as mean and standard deviation (SD) measures. The Shapiro-Wilk test was performed to check the distribution of data. Most showed normal distribution, so the parametric test was used. The demographic data were tested for significant differences between the two groups using the independent t-test. Two-way mixed measure ANOVA was used to evaluate the effect of the treatment factor (between-group), time factor (within-group), and their interactions on all dependent variables. Comparison of lumbar segmental motion in sagittal translation and sagittal rotation, plus the trunk muscle activation pattern, were measured at three time points: baseline, after the 10th week of exercises, and at 12 months follow-up. They were calculated using the one-way repeated measures ANOVA with the least significant difference (LSD) for post hoc analysis. Pain intensity was analyzed using the Wilcoxon Signed-Rank test. The mean differences between the CSE with ADIM and STE programs were determined using the analysis of covariance (ANCOVA), adjusting for all baseline outcomes. The significance level (*p*-value) was set at 0.05. All analyses were performed using SPSS version 19.0 software (SPSS Inc., Chicago, IL, USA).

## 3. Results

### 3.1. Pain Intensity

The average pain intensity within a 24-h period revealed a significant decrease in both the CSE with ADIM group (*p*-value < 0.01) and STE group (*p*-value < 0.01) at the 10 weeks assessment. However, the STE group was significantly higher than the CSE with ADIM group (*p*-value = 0.001) at the 10-week assessment. At the 12-month follow-up, the pain intensity in the CSE with ADIM group was significantly increased, but it was not significantly different from the STE group (Table 2).

### 3.2. Trunk Muscle Activity

Trunk muscle activity at each time of measurement between the two exercises is shown in Figure 3. The CSE with ADIM demonstrated significantly increased activity in the TrA and IO muscles for the 10 weeks of the exercise when compared with the baseline (*p*-value ≤ 0.05) and decreased significant activity in the TrA and IO muscles at the 12 months follow-up when compared with the 10 weeks of the exercise (*p*-value ≤ 0.05). For LM muscle, the CSE with ADIM and STE showed significantly increased activity for the 10 weeks and 12 months follow-up of the exercise when compared with baseline (*p*-value ≤ 0.05) but there was no significant difference between measures at 10 weeks and 12 months. For the ICLT muscle, the CSE with ADIM showed significantly decreased activity following 10 weeks of the exercise when compared with the baseline (*p*-value ≤ 0.05).

Comparison between the exercises indicated that the STE demonstrated significantly higher muscle activity than in the CSE with ADIM in the RA muscle at 10 weeks of exercise (*p*-value ≤ 0.01).

### 3.3. Trunk Muscle Ratio Activity

This study demonstrated the trunk muscle activation pattern in terms of the ratio of abdominal muscle (TrA and IO/RA) and the ratio of back muscle (LM/ICLT) activation. The interaction effects of abdominal muscle ratio activity were significant (*p*-value < 0.01), but the interaction effects of back muscle ratio activity were not significant (*p*-value > 0.01). The results showed significantly increased ratios of both deep abdominal (*p*-value < 0.01) and deep back muscle activation (*p*-value < 0.05) after regularly completing CSE with ADIM for 10 weeks (Table 2). The ratio of abdominal muscle was significantly maintained at the 12-month follow-up (*p*-value < 0.05) in the CSE with ADIM group.

The comparison between CSE with ADIM and STE groups on the ratio of abdominal muscle activation showed a significant difference at both 10 weeks (*p*-value = 0.002) and 12 months follow-up (*p*-value = 0.007), while the ratio of back muscle activation showed a non-significant difference between groups. However, the CSE with the ADIM group demonstrated a greater improvement of the deep trunk muscle activation pattern than the STE group (Table 2).

### 3.4. Lumbar Segmental Motion

The interaction effects (treatment × time) of all lumbar segmental motion variables were not significant (*p*-value > 0.05). Within-group measurements of sagittal translation and rotation of lumbar segments are presented in Table 3. After 10 weeks of the CSE with ADIM, there was a significantly decreased translation of L4-L5 (*p*-value < 0.01) and L5-S1 (*p*-value < 0.05), while the rotation of all segments showed a non-significant difference. Increased translation at L3-L4 (*p*-value < 0.05) and L4-5 (*p*-value < 0.05) were found to be significant at the 12 month follow-up. In the STE group, the results revealed slightly increased sagittal translation of all segments after 10 weeks of the exercise and at the 12-month follow-up. Sagittal rotation of lumbar segments at L3-L4 (*p*-value < 0.05), L4-L5 (*p*-value < 0.05), and L5-S1 (*p*-value < 0.01) were significantly increased after 10 weeks of the exercise (Table 3).

The lumbar segmental translation at L4-L5 (*p*-value = 0.001) and L5-S1 (*p*-value = 0.041) between the CSE with ADIM and STE groups after 10 weeks of exercise revealed significant differences, and the segmental rotation revealed a significant difference at the L4-5 level (*p*-value = 0.03). However, there was no difference in either translation or rotation between the CSE with ADIM and STE groups at the 12-month follow-up, as shown in Table 3.

## 4. Discussion

Both exercise groups significantly reduced the baseline pain intensity. Trunk muscle activation pattern, the ratio of abdominal muscle (TrA and IO/RA), and the ratio of back muscle (LM/ICLT) revealed improvement after 10 weeks of the CSE with the ADIM group. The results revealed that 10 weeks of CSE with ADIM significantly reduced lumbar segmental translation motion, whereas the STE demonstrated slightly increased translation and significantly increased rotation motion. However, the level of lumbar sagittal translation had a return effect at the 12-month follow-up assessment in CSE with ADIM.

The findings revealed that both the CSE with ADIM and STE groups had significantly reduced pain intensity after 10 weeks. Many studies have demonstrated that CSE with ADIM and STE can significantly reduce pain by improving trunk muscle ability in terms of coordination and recruitment patterns [50,51,52]. The current study demonstrated that pain intensity in the CSE with ADIM group showed a greater trend of improvement than the STE group. The reduction in pain intensity might be related to deep abdominal muscle activity. This is aligned with previous research that reported reduced low back pain was associated with an increased TrA contraction ratio [53]. Both exercises can reduce low back pain scores beyond a minimal clinically significant difference (MCID), which for the numeric rating scale (NRS) is 2.5 according to Ostelo and de Vet in 2005 [54].

The current study demonstrated that the CSE with ADIM group exhibited reduced lumbar segmental motion in full trunk flexion and full trunk extension, but STE increased it. One possible explanation is that the CSE with ADIM group could increase deep trunk muscle function, coactivation of TrA and IO/LM [45,55,56], and improved neuromuscular control [5,57,58]. All participants in the CSE with ADIM achieved the autonomous phase that represented increased neuromuscular control. Our study demonstrated that CSE with ADIM significantly increased the activation ratio of the abdominal muscles (TrA and IO/RA) and deep back (LM/ICLT) muscle activation (Table 2), with significantly increased activation of the TrA&IO and LM muscles after 10 weeks of the exercise when compared with the baseline (Figure 3). The results showed that the mean difference of lumbar segmental translation in the current study was higher than 0.5 mm, which was the average difference reported by Iguchi et al. in 2003 [59]. This indicated that the reduction of segmental translation after CSE with ADIM was influenced by the actual effect of our intervention. This reduces lumbar translation motion, which may have been due to a specific training effect of the CSE with ADIM, which focuses on the preferential retraining of the deep trunk muscle. The results support previous studies that have demonstrated that CSE with ADIM improves deep abdominal muscle activation [18,25], which could eventually reduce lumbar segmental motion in full trunk flexion and full trunk extension (Table 3). These may be referred that the results of lumbar segmental motion and muscle activation appear to direction as we expected (Figure 3 and Table 3). Some level of lumbar sagittal translation returned at the 12-month follow-up assessment because the participants discontinued their exercise program, reducing deep trunk muscle function; for this reason, it is not surprising that the translation of lumbar segmental motion was increased after the 12-month follow-up.

On the other hand, the STE showed significantly increased segmental translation and rotation when compared with the CSE with ADIM. The results of the STE demonstrated significantly increased RA muscle and a non-significant difference in trunk muscle activation pattern when compared with CSE with ADIM, which could imply that STE did not activate the deep abdominal muscles (Table 2). Therefore, this finding possibly arises from increased lumbar segmental motion. This supports the findings of Mohammadimajd et al. in 2020 from their study where patients with grade-I spondylolisthesis received STE for 8 weeks [60]. They found that STE could reduce pain and functional disability in patients with spondylolisthesis. Our study demonstrated that STE can reduce pain. Exercise intervention could activate endogenous pain inhibitory mechanisms and lead to a reduction in sensitivity to noxious stimuli resulting in reduced pain intensity [61]. However, STE was focused on the superficial trunk muscles that indirectly attach to the lumbar spine [25,32]. They are large and can generate high torque [25,32], which is a cause of increased segmental translation motion [5].

This study was carefully designed and controlled, although it has some limitations. The first limitation is the small sample size. Although the sample size was small (*n* = 17 per group), the effect size in the present study between treatment and control groups was 1.313, which demonstrates statistically significant effects between groups. The second is the wide age range of participants, which means that participants may have had a varied state of disc degeneration. Future studies could help improve the strength of results by using an MRI assessment to determine the stage of disc degeneration. The third is that the current study had only two groups of exercise (CSE with ADIM and STE). In future studies, one could consider having a control group to further strengthen the results. The sEMG signals may be interfered with by crosstalk from other superficial muscles, especially for the LM muscles. Further studies may consider using indwelling EMG signals to accurately measure individual muscle activation. Our study was not balanced with the number of males and females in each group and it is possible that the menstrual cycle may affect participants’ sensibility. There have been no reported MCID of muscle activity and segmental motion in previous studies. Therefore, we cannot actually interpret if exercise interventions were achieved clinically significant. Our program of exercise was only 10-weeks, unlike most previous studies which commonly used 12-weeks of exercise; it would be useful for a further study to explore the effect of the two exercises over 12-weeks of training. Synergistic coactivation of agonist and antagonist muscles is an important concept associated with dynamic stability. Further study should emphasize the quality of the EMG profile to show whether muscle activation occurs in a coordinated manner by assessing the Ensemble-average patterns for each muscle during exercise [27]. This method was suggested by Hubley-Kozey and Vezina in 2002. The last, our study participants have not reported any adverse effects and almost all of them have a mild to moderate pain score. Although some of the participants felt discomfort after an exercise session, they keep exercising throughout the program.

## 5. Conclusions

A 10-week CSE with ADIM improved the deep trunk muscle activation pattern and reduced pain intensity in CLBP patients with CLI. This exercise program can reduce the translation motion of the lumbar segments. The STE demonstrated only reduced pain intensity, and the lumbar sagittal translation was greater. At the 12-month follow-up, the level of lumbar sagittal translation was increased in both groups, which implies that patients with CLI need to continue to perform CSE with ADIM beyond the 10-week exercise program to prevent excessive segmental motion in the longer term.

## Figures and Tables

**Figure 1 ijerph-18-07811-f001:**
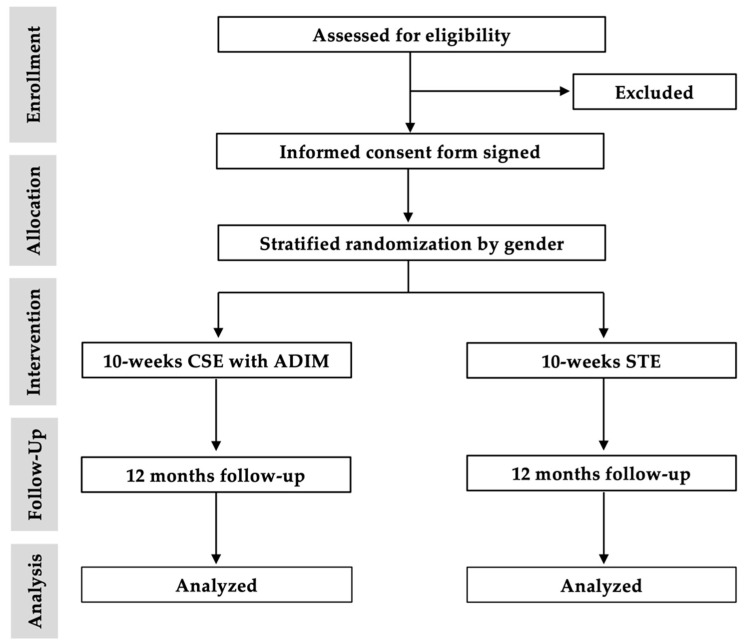
Flow chart of participants through the study.

**Figure 2 ijerph-18-07811-f002:**
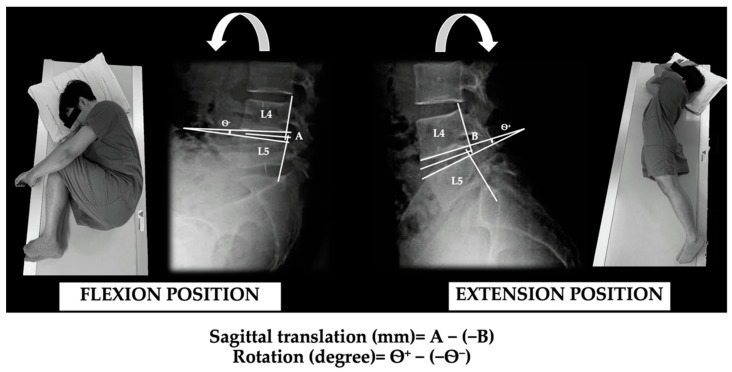
Measurement technique and positions of sagittal angulation and translation.

**Figure 3 ijerph-18-07811-f003:**
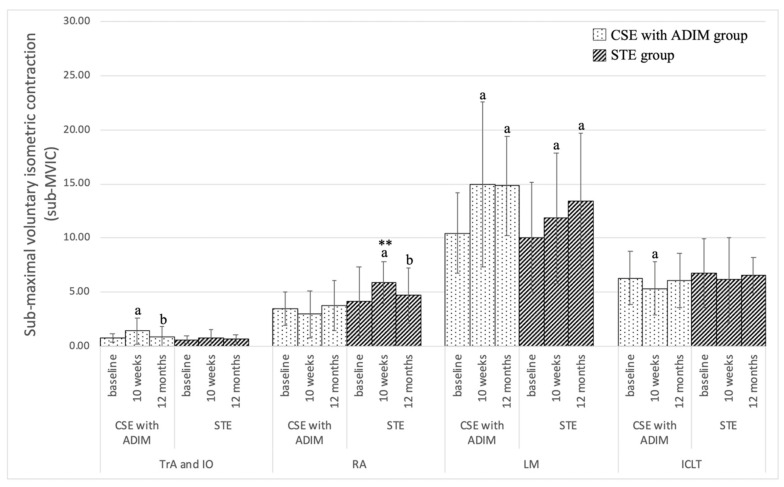
Trunk muscle activity at each time of measurement between the two exercises. Note: a = *p*-value < 0.05, difference between baseline; b = *p*-value < 0.05, difference between 10 weeks; ** = *p*-value < 0.05, difference between CSE with ADIM and STE group.

**Table 1 ijerph-18-07811-t001:** Characteristics of participants in the CSE with ADIM and the STE groups.

Characteristics	CSE with ADIM Group (*n* = 17)	STE Group (*n* = 17)
Gender		
Male	4	4
Female	13	13
Age (years)	37.29 ± 14.26	37.53 ± 11.67
Body mass (kg)	56.88 ± 8.94	57.76 ± 9.85
Stature (cm)	160.65 ± 6.98	162.12 ± 6.41
Body mass index (kg/m^2^)	22.00 ± 7.52	21.89 ± 2.83
Pain intensity	5.71 ± 1.26	5.77 ± 1.64
Education		
Undergraduate	11	12
Graduate	6	5
Career		
Student	8	4
Employee	3	6
Farmer	6	7

Notes: The data are presented using mean ± standard deviation. Abbreviations: cm = centimeter; kg = kilogram; m^2^ = square meter.

**Table 2 ijerph-18-07811-t002:** Trunk muscle activation pattern and pain intensity at 10-week and 12 months of the CSE with ADIM and the STE groups.

Variables	CSE with ADIM Group (*n* = 17)			STE Group (*n* = 17)			Adjusted Group Differences in Mean Change at 10 Weeks			Adjusted Group Differences in Mean Change at 12 Months		
	Baseline	10 Weeks	12 Months	Baseline	10 Weeks	12 Months	Mean	95% CI	*p*-Value	Mean	95% CI	*p*-Value
Trunk muscle activation patterns												
Abdominal muscle activity ratio	0.22 ± 0.14	0.48 ± 0.35 ^††^	0.23 ± 0.13 ^#^	0.15 ± 0.08	0.13 ± 0.14	0.13 ± 0.09	0.26	0.10–0.41	0.002 **	0.12	0.03–0.20	0.007 **
Back muscle activity ratio	1.64 ± 1.00	2.79 ± 1.96 ^†^	2.42 ± 1.35	1.47 ± 0.67	1.93 ± 1.45	2.04 ± 1.57	0.85	−0.38–2.08	0.170	0.38	−0.67–1.42	0.471
Pain												
Pain intensity (score)	5.71 ± 1.26	1.41 ± 1.28 ^††^	2.65 ± 1.69 ^##^	5.77 ± 1.64	2.59 ± 1.62 ^††^	3.29 ± 2.26	−1.13	−1.78–(−0.48)	0.001 **	−0.61	−1.84–0.62	0.322

Notes: The data are presented using mean ± standard deviation. Adjusted group differences in mean change were adjusted by each variable assessed at baseline. ^††^ *p*-value < 0.01 difference between baseline and 10 weeks, ^†^ *p*-value < 0.05 difference between baseline and 10 weeks. ^##^ *p*-value < 0.01 difference between baseline and 12 months, ^#^ *p*-value < 0.05 difference between baseline and 12 months. ** *p*-value < 0.01 difference in mean between the CSE with ADIM and the STE groups.

**Table 3 ijerph-18-07811-t003:** Lumbar segmental translation and rotation motion at 10-weeks and 12 months in the CSE with ADIM and the STE groups.

Lumbar Segmental	CSE with ADIM Group (*n* = 17)			STE Group (*n* = 17)			Adjusted Group Differences in Mean Change at 10 Weeks			Adjusted Group Differences in Mean Change at 12 Months		
	Baseline	10 Weeks	12 Months	Baseline	10 Weeks	12 Months	Mean	95% CI	*p*-Value	Mean	95% CI	*p*-Value
Sagittal translation, millimeters												
L3-L4	2.59 ± 0.79	2.18 ± 0.75	2.92 ± 1.03 ^#^	2.03 ± 1.15	2.43 ± 1.13	2.32 ± 1.27	−0.54	−1.14–0.07	0.081	0.15	−0.48–0.79	0.626
L4-L5	3.22 ± 1.56	2.24 ± 1.35 ^††^	3.31 ± 1.60 ^#^	2.96 ± 1.03	3.06 ± 1.11	3.20 ± 1.71	−1.01	−1.58–(−0.44)	0.001 **	−0.04	−1.10–1.02	0.936
L5-S1	3.61 ± 2.52	3.02 ± 2.64 ^†^	4.13 ± 3.07	2.38 ± 1.17	2.81 ± 1.52	4.67 ± 3.79	−0.93	−1.82–(−0.44)	0.041 *	−1.14	−3.62–1.33	0.353
Sagittal rotation, degrees												
L3-L4	15.88 ± 4.69	15.35 ± 3.32	14.36 ± 2.94	13.59 ± 4.24	15.35 ± 3.81 ^†^	14.56 ± 4.25	−1.41	−3.09–0.27	0.096	−0.70	−3.29–1.88	0.583
L4-L5	18.65 ± 4.70	17.94 ± 5.31	16.19 ± 4.42	17.00 ± 3.20	19.59 ± 4.21 ^†^	18.11 ± 4.99	−2.93	−5.56–(−0.31)	0.030 *	−2.69	−5.83–0.46	0.091
L5-S1	21.47 ± 8.06	22.53 ± 9.89	22.29 ± 8.19	23.65 ± 6.70	26.47 ± 8.07 ^††^	22.76 ± 11.57	−1.65	−4.94–(1.63)	0.313	0.93	−5.40–7.27	0.766

Notes: The data are presented using mean ± standard deviation. Adjusted group differences in mean change were adjusted by each variable assessed at baseline. ^††^ *p*-value < 0.01 difference between baseline and 10 weeks, ^†^
*p*-value < 0.05 difference between baseline and 10 weeks. ^#^ *p*-value < 0.05 difference between baseline and 12 months. ** *p*-value < 0.01 difference in mean between the CSE with ADIM and the STE groups, * *p*-value < 0.05 difference in mean between the CSE with ADIM and the STE groups.

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
