# Peer review of "The Effects of Core Stabilization Exercise with the Abdominal Drawing-in Maneuver Technique versus General Strengthening Exercise on Lumbar Segmental Motion in Patients with Clinical Lumbar Instability: A Randomized Controlled Trial with 12-Month Follow-Up"

_ijerph, 2021, doi:10.3390/ijerph18157811_

Round 1

Reviewer 1 Report

The manuscript was designed to compare the effect of 10 weeks of core stabilization exercises (CSE) using the abdominal drawing-in maneuver technique (ADIM) and general strengthening exercise (STE). It was also aimed to follow-up 12 months on lumbar segmental motion (L3-L4, L4-L5, and L5-S1), trunk muscle activation, and pain intensity in CLBP patients with clinical lumbar instability (CLI).

The introduction is well structured and presents the problem and the research question. However, minor points require adjustment. First, trunk exercises aimed to improve lumbar stability have been used for a long time. As suggested, they were not applied recently, as indicated – see your reference 50, which provides relevant background arguments. I would also recommend authors to refer to the work by Hubley-Kozey and colleagues (Hubley-Kozey CL, Vezina MJ. Muscle activation during exercises to improve trunk stability in men with low back pain. Arch Phys Med Rehabil 2002;83:1100-8) and by Davidson and colleagues (Clarke Davidson KL, Hubley-Kozey CL. Trunk muscle responses to demands of an exercise progression to improve dynamic spinal stability. Arch Phys Med Rehabil 2005;86:216-23).

One relevant question posed in the introduction is related to the study performed by Javadian and colleagues (2015), in which CSE + ADIM and STE were effective, with additional benefits for the first one.  It must be more explicit the advances may be provided by including EMG analysis. At least a perspective of the benefits EMG use must be provided to improve the study's rationale. As it stands, the difference is that no one has done it before in this context. What can EMG reveal? Please, observe the work performed by Hubley-Kozey and colleagues as they have performed a comparable analysis (i.e., measured the EMG of several stabilizing muscles). In addition, it concerns the fact that such muscles are deep, which makes EMG measures difficult and sometimes problematic.

Methods

Why were only ten weeks performed as most programs involve 12 weeks? Please justify.

The age range is quite extensive, and some of them may also have other issues (disk degeneration), which must be controlled, or it may become an important confounding factor. It is also relevant to identify the way other spinal disorders were identified (how were they determined?)

Please, move (and expand details) of the consort figure to the methods section to illustrate your allocation strategy and procedures. Note that no results are reported but that it is impossible to identify how many subjects were recruited and discarded (by cause) when your criteria were set in place.

It is mentioned that exercise progression was adjusted, but it is too vague. It is not possible to identify how or what parameters were controlled to perform the exercise progression. Does it bring other questions as to how the biofeedback was provided across the training sessions? How did participants manage that? The same applies to the EMG. Prescribing by time is something complicated, as no number of repetitions and series were provided. Please, develop further on this as the prescription is somewhat confusing.

As far as I got the idea, you have two groups and no proper control group. I would not do differently as improving the experimental group is preferable to having a “pure control.” However, it is an issue that must be addressed.

It is also interesting that you have twice women as men. Please, address the fact that the menstrual cycle is an uncontrolled intervening factor.

Results

Please replace weight (a measure of force) with body mass (expressed in Kg – a mass measurement). Also, replace “height” with “stature”.

Operationally the acronym CSE with ADIM is far too big. It may be simplified.

The EMG is poorly described, and such critical parameter requires further details. Collecting EMG data from deep muscles is not simple, and a clear protocol must be provided to increase the confidence in your data. Note that there is no minimal description of the procedures. There is neither mention of where the electrodes were positioned nor the control for crosstalk. How confident are authors that EMG traces are collected from the selected muscles authors intended to analyze?

Conceptually, please, clarify how the ratio is a good indication of muscle recruitment as it considers two separate muscle groups? It reveals what the pair of agonists and antagonists are doing. The ratio could be preserved (unchanged) if both muscle groups increase activation. Note that the rate can not reveal increasing activation! Besides, if the ratio increased, one group likely increased its activation over the other. Nothing is commented regarding this. Likewise, sustaining the ratio in the following may result from comparable decreases of extensor and flexor muscle groups. Therefore, I insist, the ratio can not assess the activation pattern.

It is contrasting the main argument to perform the study (EMG) to be so underreported.

It would be handy to have a minimal detectable change (MDC) measure. I understand the intra-rater reliability. I suspect only one experimenter assessed the data. Hence, intra rater is of little use compared to the ability of a single examiner to detect clinically significant changes (MDC).

Regarding your stats, I would like to invite authors to check if the normality assumptions were met. Otherwise, the parametric approach may not be applicable.

The first paragraph is confusing as it mentions 34 participants entered the study, then the exclusion criteria were applied, which resulted in two groups of 17 (2x17=34). The criteria were used on a different group (of 65 individuals). Please reorganize.

Table 2 is very confusing and cluttered. Please, reorganize.

The reduced displacement of L4-L5 may be assumed after the MDC is provided.

Please, be more specific regarding interactions as both “treatments” resulted in improved stability.

I argue and advise against the use of the ratio as a clear indicator or muscle activation pattern. In addition, it is assumed that such changes (in muscle activation) are related to increased stability. Hummmm!!! If participants freeze (due to pain or fear of pain), they would also reduce the range of motion and mobility.

The quote that increases deep abdominal and back muscle activation is equivocal. If both increase by the same rate, the ratio remains the same. Then, mentioning that one protocol improves deep abdominal muscle activation requires further clarification as it can not be derived from the ratio.

Please, clarify why increased translation matches with the activation rates in the STE group. It sounds opposite to the rationale applied in your study. Please, clarify.

Author Response

Date 11 July 2021

Subject: Manuscript re-submission ijerph-1277146

Dear Reviewer 1

I would like to re-submit the manuscript titled “ The Effects of Core Stabilization Exercise with the Abdominal Drawing-in Maneuver Technique Versus General Strengthening Exercise on Lumbar Segmental Motion in Patients with Clinical Lumbar Instability: A Randomized Controlled Trial with 12-Month Follow-up” for publication in the International Journal of Environmental Research and Public Health.

This study compared outcomes in 34 lumbar instability patients in two exercises at 10 weeks and 12 months follow up. Participants were divided into either Core stabilization (deep) exercise, incorporating abdominal drawing-in maneuver technique (CSE with ADIM), or General strengthening (superficial) exercise (STE). Outcome measures werepain, muscle activation and lumbar segmental motion. Participants in CSE with ADIM had significantly less pain than those in STE at 10 weeks. They showed significantly more improvement of abdominal muscle activity ratio than participants in STE at 10 weeks and 12 months follow-up. Participants in CSE with ADIM had significantly reduced sagittal translation at L4-L5 and L5-S1 compared with STE at 10 weeks. Participants in CSE with ADIM had significantly reduced sagittal translations at L4-L5 and L5-S1 compared with participants in STE at 10 weeks, whereas STE demonstrated significantly increased sagittal rotation at L4-L5. However, at 12 months follow-up, levels of lumbar sagittal translation were increased in both groups. CSE with ADIM which focuses on increasing deep trunk muscle activity can reduce lumbar segmental translation and should be recommended for lumbar instability.

All authors have contributed significantly to the manuscript. We thank the editor and reviewers for their time in evaluating our work. We have responded to all concerns raised on a point-by-point basis. We confirm that the article is original, that it is not under consideration by other journals, and that this material has not been previously published.

Thank you for your time and attention, please contact us if we can provide you with additional changes or material

Sincerely Yours,

Assoc. Prof. Dr. Rungthip Puntumetakul

Faculty of Associated Medical Science,

Khon Kaen University, Khon Kaen 40002, Thailand

Reviewer 2 Report

The Effects of Core Stabilization Exercise with the Abdominal Drawing in Maneuver Technique Versus General Strengthening Exercise on Lumbar Segmental Motion in Patients with Clinical Lumbar Instability: A Randomized Controlled Trial with 12-Month Follow-up

This trial compared outcomes in 34 lumbar instability in two exercises at 10 weeks and 12 months follow up. Participants were divided into either Core stabilization (deep) exercise, incorporating abdominal drawing-in maneuver technique (CSE with ADIM), or General strengthening (superficial) exercise (STE). A previous study has compared the same intervention for the same outcome over eight weeks. This study explored the long-term effect of the treatment at 10 weeks and 12 months follow-up. Additionally, the used electromyography (EMG) measures to assess trunk muscle activity.

Comments:

Participants were selected by aberrant movement sign, painful catch sign and prone instability. Positive test results on two out of the three tests led to a diagnosis of lumbar instability by the physical therapist. Patients with other disorders were excluded. Please give a detailed list of the number of people screened and excluded in each category.

Please give more details on the sample size calculation. What was the primary outcome for which sample size was calculated? What was the standard deviation? Effect size estimated?

The definition of outcomes of the study is confusing. Do you have 3 co-primary outcomes? Please specify which was the primary outcomes for which sample size was calculated. Rest all should be secondary outcomes. A hypothesis should be mentioned for the primary outcome.

It seems that the randomisation was not stratified by gender, but how come equal males and females in each group? Hard to believe it is by chance alone in such a small study

Table 2 mentions that it is adjusted. Please note the variables adjusted in the footnote and in the stats section. Do you mean that the change in the variable was only adjusted for the baseline variable? Not age, sex, BMI?

You can also use the repeated-measures mixed-effects model to account for the slope of the overall change in outcome over 12 months.

Can you discuss the effect size you observed against the minimum clinically important difference in back pain? I can see that the differences in pain intensity are above MCID. Do you have similar data from previous literature on muscle ratio, sagittal rotation and translation?

All clinical trials should have collected Adverse events. Can you add a table with the list of adverse events (even if they are not listed as related to intervention)

Author Response

Date 11 July 2021

Subject: Manuscript re-submission ijerph-1277146

Dear Reviewer 2

I would like to re-submit the manuscript titled “ The Effects of Core Stabilization Exercise with the Abdominal Drawing-in Maneuver Technique Versus General Strengthening Exercise on Lumbar Segmental Motion in Patients with Clinical Lumbar Instability: A Randomized Controlled Trial with 12-Month Follow-up” for publication in the International Journal of Environmental Research and Public Health.

This study compared outcomes in 34 lumbar instability patients in two exercises at 10 weeks and 12 months follow up. Participants were divided into either Core stabilization (deep) exercise, incorporating abdominal drawing-in maneuver technique (CSE with ADIM), or General strengthening (superficial) exercise (STE). Outcome measures werepain, muscle activation and lumbar segmental motion. Participants in CSE with ADIM had significantly less pain than those in STE at 10 weeks. They showed significantly more improvement of abdominal muscle activity ratio than participants in STE at 10 weeks and 12 months follow-up. Participants in CSE with ADIM had significantly reduced sagittal translation at L4-L5 and L5-S1 compared with STE at 10 weeks. Participants in CSE with ADIM had significantly reduced sagittal translations at L4-L5 and L5-S1 compared with participants in STE at 10 weeks, whereas STE demonstrated significantly increased sagittal rotation at L4-L5. However, at 12 months follow-up, levels of lumbar sagittal translation were increased in both groups. CSE with ADIM which focuses on increasing deep trunk muscle activity can reduce lumbar segmental translation and should be recommended for lumbar instability.

All authors have contributed significantly to the manuscript. We thank the editor and reviewers for their time in evaluating our work. We have responded to all concerns raised on a point-by-point basis. We confirm that the article is original, that it is not under consideration by other journals, and that this material has not been previously published.

Thank you for your time and attention, please contact us if we can provide you with additional changes or material

Sincerely Yours,

Assoc. Prof. Dr. Rungthip Puntumetakul

Faculty of Associated Medical Science,

Khon Kaen University, Khon Kaen 40002, Thailand

Reviewer 3 Report

This study mainly conducted a core stabilization exercise with the abdominal drawing-in maneuver technique (CSE with ADIM) at a 12-month follow-up and compared with the traditional strengthening exercise (STE) for 34 lumbar segmental motion in patients with clinical lumbar instability. The differences in the related measurements were also examined when performing the maneuver techniques after a period of 10-week. This manuscript is generally well written and practical valuable. However, several issues I concern should be corrected or further clarified prior to its publication in ijerph.

 Ln26, What did “34 lumbar instability” mean? Did it mean the patients?

Ln31, abdominal muscle ratio. I suggest that “abdominal muscle activity ratio” may be the better term. Please check it through the manuscript.

Ln30-34, For results description, this sentence is too long to be easily understood. Please separated the sentence into 2 or 3 sentences to enhance the clearness.

L34-35, “While STE demonstrated significantly increased sagittal rotation of 34 lumbar segmental motion at L4-L5 (p-value = 0.03) compared with CSE with ADIM at 10 weeks.” This sentence is incomplete.

Ln68-75, The authors should explain why the study was conducted and how about its contribution because the benefits of the CSE with ADIM combined with the STE on the related patients had been verified by Javadian et al. in 2015. Only the longer follow-up period for these exercises may not be enough. More explanations are needed.

Ln81-82, three measurement points

Ln87, Please give the study duration.

Ln181-182, the reference(s) regarding the method of submaximal voluntary isometric contractions for back muscles should be provided, as for the abdominal muscles.

Ln206, More information is needed about what statistical analysis software was used in the study.

Ln212, one-way repeated ANOVA with LSD for post hoc analysis. What authors meant here may be “one-way repeated-measures ANOVA”. In addition, please spell out the LSD and then use the abbreviation later.

Ln217, The authors could consider to move subsection 3.1 Participant characteristics, as well as Figure 2 and Table 2, to subsection 2.2.

Ln253, Table 2, The 95% CI can be presented as 0.10-0.41, the term “to” should be removed and the rest may be inferred by analogy.

Ln68, 101, 158, 296, 317, The references were cited by an incorrect format.

Author Response

Date 11 July 2021

Subject: Manuscript re-submission ijerph-1277146

Dear Reviewer 3

I would like to re-submit the manuscript titled “ The Effects of Core Stabilization Exercise with the Abdominal Drawing-in Maneuver Technique Versus General Strengthening Exercise on Lumbar Segmental Motion in Patients with Clinical Lumbar Instability: A Randomized Controlled Trial with 12-Month Follow-up” for publication in the International Journal of Environmental Research and Public Health.

This study compared outcomes in 34 lumbar instability patients in two exercises at 10 weeks and 12 months follow up. Participants were divided into either Core stabilization (deep) exercise, incorporating abdominal drawing-in maneuver technique (CSE with ADIM), or General strengthening (superficial) exercise (STE). Outcome measures werepain, muscle activation and lumbar segmental motion. Participants in CSE with ADIM had significantly less pain than those in STE at 10 weeks. They showed significantly more improvement of abdominal muscle activity ratio than participants in STE at 10 weeks and 12 months follow-up. Participants in CSE with ADIM had significantly reduced sagittal translation at L4-L5 and L5-S1 compared with STE at 10 weeks. Participants in CSE with ADIM had significantly reduced sagittal translations at L4-L5 and L5-S1 compared with participants in STE at 10 weeks, whereas STE demonstrated significantly increased sagittal rotation at L4-L5. However, at 12 months follow-up, levels of lumbar sagittal translation were increased in both groups. CSE with ADIM which focuses on increasing deep trunk muscle activity can reduce lumbar segmental translation and should be recommended for lumbar instability.

All authors have contributed significantly to the manuscript. We thank the editor and reviewers for their time in evaluating our work. We have responded to all concerns raised on a point-by-point basis. We confirm that the article is original, that it is not under consideration by other journals, and that this material has not been previously published.

Thank you for your time and attention, please contact us if we can provide you with additional changes or material

Sincerely Yours,

Assoc. Prof. Dr. Rungthip Puntumetakul

Faculty of Associated Medical Science,

Khon Kaen University, Khon Kaen 40002, Thailand

Round 2

Reviewer 1 Report

Initially, I congratulate authors for their efforts in replying and rebating the criticism raised in the first round. Indeed, the manuscript has improved in several aspects, but there is margin for improvement as some points were clarified and others were not fully answered. In this round I will address only to the relevant points.

Point 1:

Response 1: Thank you for your suggestion. We have rewritten as follows:

 “Trunk stability exercises have been recommended for patients with CLI to decrease pain, reduce disability, improve quality of life, trunk muscle function, and quality of lumbar segmental motion [5, 18-23].” 

Line: 59-61

REVIEWER: The idea here was to invite authors to think that coordination, i.e., the activation timing may be relevant rather than the amount of activation. Thus, the amount the muscles are recruited may be less important than the actual organization in which muscles are used to stabilize the motion segment. Hubley-Kozey and colleagues showed that low amounts of activation may produce positive outcomes. Please, consider these arguments rather than only focusing on the amounts and ratios of activation. It opposes the arguments presented by the authors and follows the same directions of the criticism observed on the point 2 (

Point 2: One relevant question posed in the introduction is related to the study performed by Javadian and colleagues (2015)....

Response 2:

This was the most commonly used reference when attempting to establish a physiologic measure related to the recruitment capability of a muscle [28].”

Line: 75-79

REVIEWER: Note here the same argument regarding organization rather than the amount of muscle recruitment (RMS). No comments regarding deep muscles were included. Javadian’s work did not assess EMG, but others did. Again, a clear rationale is still missing (see Puntumetakul et al.).

Point 3: Methods

Why were only ten weeks performed as most programs involve 12 weeks? Please justify.

Response 3: They found 10-weeks of CSE with ADIM were sufficient to improve deep trunk muscle activity, although this differs from most programs which involve 12-weeks of exercise, as you mention.

REVIEWER: Fine, but please remember that a 12 period would be more relevant, and it adds to the list of limitations that must be acknowledged. The explanation to the reviewer was delivered, now, please, make sure it is justified for the reader. It must be explained in the text.

Point 4: The age range is quite extensive..

Response 4:

 “The second is the wide age range..

REVIEWER: Fine…

Point 5: Please, move (and expand details) of the consort figure …

Response 5: Thank you for your suggestion. We have added more details to the consort figure and moved it into the Methods section.

REVIEWER: Fine…

Point 6: It is mentioned that exercise progression …

Response 6: Thank you for your suggestion. We have explained this section in more detail as follows:

“Details of the CSE …

Line: 167-196

REVIEWER: Fine

Point 7: As far as I got the idea…

Response 7: Thank you for your suggestion and we agree…

Line: 425-427

REVIEWER: Fine

Point 8: It is also interesting that you have twice women as men. Please, address the fact that the menstrual cycle is an uncontrolled intervening factor.

Response 8: Thank you for your comment….

REVIEWER: I am not referring between groups. Within groups it is unbalanced. Menstrual cycle may impact on your patients’ sensibility. I kindly ask authors to expand limitations regarding this issue.

Point 9: Results

Please replace weight…

Response 9: We have changed …

REVIEWER: Fine

Point 10: Operationally the acronym…

Response 10: We understand of your concern ...

REVIEWER: Fine

Point 11: The EMG is poorly described…

Response 11: We have revised details of the EMG as follows:

The ratio muscle activation of mean RMS value (mean RMS of deep muscle divided by mean RMS of superficial muscle) has previously been used to represent the change in muscle activation [45, 46].

REVIEWER: Methods sounds more robust! The ration does not express muscle activation! It’s a ratio. It reveals the ratio between a muscle pair. Please, adjust. Once more I insist that increasing the activation of the pair will result in the same ratio. No comments regarding how cross talk was performed are provided. From my point of view, this is another limitation that must be included – with not impact on the academic merit of the manuscript.

Point 12: Conceptually, please, …

Response 12: The trunk muscle activity ratio value …

REVIEWER: Remember you are describing changes (trade-offs) between agonist and antagonist muscles.

Previous studies by Areeudomwong et al. in 2012 and Puntumetakul et al. in 2013 found that patients with CLI who receive the 10 weeks of CSE with ADIM exercise program could increase the activation ratio of the trunk muscles, with increased activation of deep muscles. Therefore, this supports the results of our study demonstrating the improvement of ratio activation of trunk muscles after 10 weeks of CSE with ADIM.

REVIEWER: Here is where the reviewer questions what is the contribution of the study, as it seems to be done before..

Point 13: It is contrasting..

Response 13: We have reported …

REVIEWER: Fine

Point 14: It would be handy…

Response 14: A previous study…

Line:394-396

REVIEWER: Fine

Point 15: Regarding your stats, I would like to invite authors to check if the normality assumptions were met. Otherwise, the parametric approach may not be applicable.

Response 15: We have re-checked our normality of each parameter. Most of them have shown normal distribution. Therefore, we decided to use parametric test.  

REVIEWER: great news… just need to report it formally in the text to assure the reader the stats is correct.

Point 16: The first paragraph…

Response 16: We have reorganized the Results section as follows:

REVIEWER: Fine

Point 17: Table 2 is very confusing and cluttered. Please, reorganize.

Response 17: Thank you for your comment. We have reorganized Table 2 and 3 by placing it at the horizontal line.

REVIEWER: Fine

Point 18: The reduced displacement of L4-L5 may be assumed after the MDC is provided.

Response 18: We have added this issue in our Discussion section as follows:

REVIEWER: Fine, although a MDC measure could be provided.

Point 19: Please, be more specific regarding interactions as both “treatments” resulted in improved stability.

Response 19: We have tried to clarify this issue in our Discussion…

REVIEWER: Fine

The STE exercise may focus on the superficial trunk muscles that indirectly attach to the lumbar spine; these muscles are large in size and can generate high torque [25, 31], which is a cause of increased trunk motion [5]. They found that STE could reduce pain and functional disability in patients with a spondylolisthesis. However, STE seemed to increase excessive lumbar vertebrae translation and rotation.

REVIEWER: it is interesting as it opposes the idea of stabilization… please comment on that.

Point 20: I argue and advise against the use of the ratio as a clear indicator or muscle activation pattern. In addition, it is assumed that such changes (in muscle activation) are related to increased stability. Hummmm!!! If participants freeze (due to pain or fear of pain), they would also reduce the range of motion and mobility.

Response 20: Thank you for your comment. As you mention, if participants freeze (due to pain or fear of pain), they would also reduce the range of motion and mobility especially during x-ray assessment.

REVIEWER: Fine, but I would comment on that in the text rather keeping between authors and the reviewer.

Point 21: The quote that increases deep abdominal and back muscle activation is equivocal. If both increase by the same rate, the ratio remains the same. Then, mentioning that one protocol improves deep abdominal muscle activation requires further clarification as it can not be derived from the ratio.

Response 21: We have added the details about %sub-MVIC in the Results section to clarify the pattern of each trunk muscle activity as follows:

Comparison between the exercises indicated that the STE demonstrated significantly higher muscle activity than in the CSE with ADIM in the RA muscle at 10 weeks of exercise (p-value ≤ 0.01).

REVIEWER: My issue is unresolved by using submaximal activation. My question refers to the use of a ratio. Once more, remember that a ration can be modified by the interaction of several muscle activation possibilities. I do not need to increase the activation of a muscle! I can change the ration decreasing the activation of its’ antagonist pair.

Point 22: Please, clarify why increased translation matches with the activation rates in the STE group. It sounds opposite to the rationale applied in your study. Please, clarify.

Response 22: we have reported the %sub-MVIC in each muscle during the time of measurements in the figure 3.

REVIEWER: The authors missed my point. Please I reinforce my previous question that was unanswered. The arguments did not address the point presented here. It means that increasing the strength, one may also increase translation? It sounds contradicting… please develop further on that and try to explain why rather than stating that someone else found the same (or similar).

“The STE exercise may focus on the superficial trunk muscles that indirectly attach to the lumbar spine”

REVIEWER: Please, add a reference for such statement.

Some new issues..

What is “quality of the motion segment”?

Author Response

Date 20th -July-2021

Subject: Manuscript re-submission ijerph-1277146

Dear Reviewer 1

I would like to re-submit the manuscript titled “ The Effects of Core Stabilization Exercise with the Abdominal Drawing-in Maneuver Technique Versus General Strengthening Exercise on Lumbar Segmental Motion in Patients with Clinical Lumbar Instability: A Randomized Controlled Trial with 12-Month Follow-up”. We have carefully edited our manuscript following each of the reviewers’ comments. The language throughout our manuscript was rechecked and revised by one of the authors, who is a native English speaker (RB). After that it was proofread by SCRIBENDI.

All authors have contributed significantly to the manuscript. We thank the editor and reviewers for their time in evaluating our work. We confirm that the article is original, that it is not under consideration by other journals, and that this material has not been previously published.

Thank you for your time and attention, please contact us if we can provide you with additional changes or material

Sincerely Yours,

Assoc. Prof. Dr. Rungthip Puntumetakul

Faculty of Associated Medical Science,

Khon Kaen University, Khon Kaen 40002, Thailand

Reviewer 2 Report

Please add the relevant responses to the limitation section. E.G. response 7: mention that there were no papers showing MCID on ----

An adverse event is any event that appears during the study (even an exaggeration of pre-existing symptom/condition). I agree that there may not be any difference between the groups regarding the Aes but its unliley that you didn't get a single AE. Please mention in the limitation if approporate

'only serious adverse events were recorded in the study and it is possible that there could be a difference  -------

Author Response

Date 20th -July-2021

Subject: Manuscript re-submission ijerph-1277146

Dear Reviewer 2

I would like to re-submit the manuscript titled “ The Effects of Core Stabilization Exercise with the Abdominal Drawing-in Maneuver Technique Versus General Strengthening Exercise on Lumbar Segmental Motion in Patients with Clinical Lumbar Instability: A Randomized Controlled Trial with 12-Month Follow-up”. We have carefully edited our manuscript following each of the reviewers’ comments. The language throughout our manuscript was rechecked and revised by one of the authors, who is a native English speaker (RB). After that it was proofread by SCRIBENDI.

All authors have contributed significantly to the manuscript. We thank the editor and reviewers for their time in evaluating our work. We confirm that the article is original, that it is not under consideration by other journals, and that this material has not been previously published.

Thank you for your time and attention, please contact us if we can provide you with additional changes or material

Sincerely Yours,

Assoc. Prof. Dr. Rungthip Puntumetakul

Faculty of Associated Medical Science,

Khon Kaen University, Khon Kaen 40002, Thailand
